# It's Never Too Late: Fusing Acoustic Information into Large Language Models for Automatic Speech Recognition

**Chen Chen**[1]* **Ruizhe Li**[2] **Yuchen Hu**[1] **Sabato Marco Siniscalchi**[3]
**Pin-Yu Chen**[4] **Eng Siong Chng**[1] **Chao-Han Huck Yang**[5]*,[6]
[1]Nanyang Technological University [2]University of Aberdeen [3]University of Palermo
[4]MIT-IBM Waston AI Lab [5]Georgia Institute of Technology [6]NVIDIA Research

## Abstract

Recent studies have successfully shown that large language models (LLMs) can be successfully used for generative error correction (GER) on top of the automatic speech recognition (ASR) output. Specifically, an LLM is utilized to carry out a direct mapping from the N-best hypotheses list generated by an ASR system to the predicted output transcription. However, despite its effectiveness, GER introduces extra *data uncertainty* since the LLM is trained without taking into account acoustic information available in the speech signal. In this work, we aim to overcome such a limitation by infusing acoustic information before generating the predicted transcription through a novel late fusion solution termed **U**ncertainty-**A**ware **D**ynamic **F**usion (UADF). UADF is a multimodal fusion approach implemented into an auto-regressive decoding process and works in two stages: (i) It first analyzes and calibrates the token-level LLM decision, and (ii) it then dynamically assimilates the information from the acoustic modality. Experimental evidence collected from various ASR tasks shows that UADF surpasses existing fusion mechanisms in several ways. It yields significant improvements in word error rate (WER) while mitigating data uncertainty issues in LLM and addressing the poor generalization relied with sole modality during fusion. We also demonstrate that UADF seamlessly adapts to audio-visual speech recognition.

## 1 Introduction

In recent years, Large Language Models (LLMs) have emerged as an epistemic beacon in the field of natural language processing (NLP), endowing text-based tasks with substantial performance gains through their extensive knowledge repository and remarkable generation capabilities (OpenAI, 2023; Anil et al., 2023; Touvron et al., 2023a;b). Additionally, the prowess of LLMs is not confined to textual data alone. They have demonstrated the capability to perceive and process non-textual information (Li et al., 2023; Lyu et al., 2023; Han et al., 2023; Wu et al., 2023b), bridging the gap between various modalities. This multifaceted understanding allows LLMs to serve as a universal interface, facilitating an intricate fusion amidst disparate data modalities for multi-modal tasks, e.g. image description (Wang et al., 2023) and speech translation (Zhang et al., 2023a).

Compared to the semantic-level fusion between images and texts, perceiving speech signals for LLMs remains a complex endeavor. The challenge stems from the inherent high sampling rate of acoustic data and the substantial modality gap that exists when transitioning between audio and textual data. Furthermore, prompting LLMs for automatic speech recognition (ASR) tasks presents a particular barrier as ASR requires learning precisely frame-level alignment instead of utterance-level understanding. Previously, the realm of ASR predominantly employed n-grams or neural language models (Yang et al., 2021a) (LMs) to rescore an N-best hypothesis list, culminating in the selection of the 1-best sentence. A recent development in this field, Chen et al. (2023b) introduce a generative error correction (GER) benchmark and prompting methods for LLM-enhanced ASR wherein the N-best hypotheses list provides informative elements to directly predict output tran-

---

*Corresponding authors: CHEN1436@e.ntu.edu.sg, hucky@nvidia.com.

scription. Notwithstanding its innovation, this methodology inadvertently introduces an enhanced level of data uncertainty, primarily because the inherent acoustic information remains agnostic during LLMs learning. This quandary is further compounded by existing fusion strategies.

Methods such as semantic token-based early fusion (Fathullah et al., 2023; Deshmukh et al., 2023) or representation post-encoding through cross-attention (termed as mid fusion) (Lei et al., 2023; Radhakrishnan et al., 2023a) offer potential solutions. However, the early fusion approach tends to bias the model's learning towards a specific modality quite easily, and the mid fusion suffers from the typically longer sequence length of speech signals compared to text sequences (Wu et al., 2023a). In addition, their efficacy is often circumscribed by inherent modality disparity. Furthermore, prematurely fusing two modalities may give rise to *modality laziness* problem (Du et al., 2023a), which describes the neural network's tendency to excessively rely on a specific modality that even multimodal performance does not surpass that of unimodal performance. This paper tries to shed light upon multimodal fusion for LLM-enhanced ASR tasks. Based on the GER-empowered Hypotheses-to-transcription (H2T) paradigm, LLMs acquire the capacity to provide an independent probability distribution based on text-only modality, allowing us to explore a fundamental fusion strategy for auto-regressive token prediction. In particular, we devise a novel framework named **U**ncertainty-**A**ware **D**ynamic **F**usion (UADF) that performs step-wise late fusion in the auto-regressive decoding process. Key to our framework, we leverage the token-level uncertainty estimation to dynamically determine the fusion weight allocated to each modality in each decoding step. This mechanism is consistent with human multimodal perception: when the primary modality is equivocal, then we spontaneously seek compensatory information from another modality.

In summary, we make several salient contributions. Firstly, we underscore the challenges inherent in fusing acoustic information into LLMs for ASR and implement several feasible fusion strategies for comparative analysis. Secondly, we introduce a novel uncertainty-aware dynamic fusion technique, UADF, which dynamically allocates modality weights in the auto-regressive decoding process, thus significantly reducing the likelihood of the occurrence of the modality laziness phenomenon. Thirdly, experimental evidence shows UADF achieves remarkable performance gain in terms of word error rate (WER), outperforming a bunch of established GER baselines. Lastly, UADF demonstrates strong generalization on other multimodal auto-regressive tasks that are seamlessly incorporated as a plug-in for audio-visual speech recognition.

## 2 RELATED WORK

**Language Modeling in ASR.** To improve linguistic acceptability, there has been considerable prior efforts in applying Language Models (LMs) within ASR system (Jelinek, 1976; Ljolje et al., 1999; Mohri et al., 2008; Sak et al., 2010; Chorowski & Jaitly, 2016; Chen et al., 2019; Hu et al., 2020; Wang et al., 2022; Liu et al., 2023a). ASR designs have been firstly explored as an acoustic model (AM) and a language model (LM), independently trained, within a noisy channel framework (Jelinek, 1976; Dixon & Silverman, 1975). The LM could be integrated in an efficient first-pass decoding (Kuo et al., 2002; Mohri et al., 2008; Liu et al., 2023b) and in second-pass rescoring to manage larger LMs (Ljolje et al., 1999; Sak et al., 2010). Despite the shift to hybrid HMM-DNN models, the basic decoding/rescoring structure persisted and led to new fusion approaches for LM integration under the emergence of end-of-end (E2E) ASR models (Chorowski & Jaitly, 2016; Sriram et al., 2017). Further advancements included a two-pass E2E ASR combining streaming and full-context decoding (Sainath et al., 2019), and a deliberation network that enhanced output generation by attending to both acoustic representations and first-pass hypotheses (Hu et al., 2020), showcasing the ongoing evolution and integration of LMs in ASR. With recent advancements in pre-trained language models (PLMs), language models have been playing versatile roles within ASR systems, e.g., bi-directional rescoring (Xu et al., 2022), knowledge distillation (Futami et al., 2020), and error correction (Leng et al., 2023; Chen et al., 2023a). More recently, Chen et al. (2023b); Yang et al. (2023); Radhakrishnan et al. (2023a) proposed an LLM-enhanced ASR benchmark called generative error correction, which learns a hypotheses-to-transcription mapping by LLMs with a LoRA adapter. In particular, since GER enables LLMs to predict transcription based on text-only modality, this work leverages this capacity and considers integrating audio modality into LLMs.

**Fusion based Multimodal Learning.** Multimodal fusion is one of the most fundamental topics in multimodal learning, which aims to integrate available modalities into a uniform learning frame-

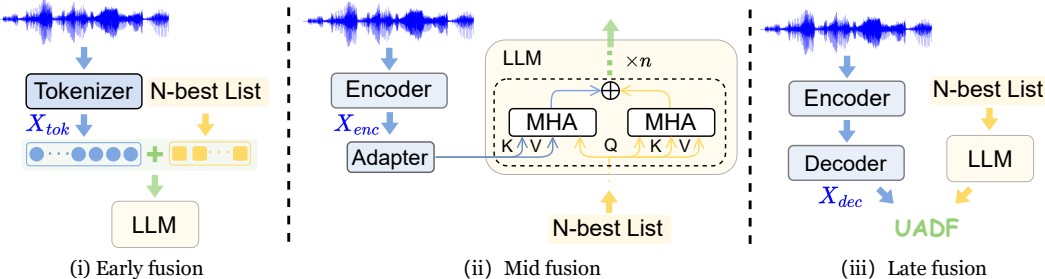

Figure 1: Different fusion strategies: early, mid and late fusions. The the green area indicates where the fusion strategies happened. *N-best* List is generated by ASR engine with beam search decoding. *Left*: the speech tokens extracted from the acoustic encoder are directly concatenated with the corresponding word embeddings of the N-best list before feeding into the LLMs; *Middle*: the acoustic features from the last layer of the acoustic encoder are integrated into the LLMs decoding process using the cross-attention mechanism; *Right*: the step-wise fusion happens in the auto-regressive decoding process by integrating both decision-level information.

work (Xu et al., 2021; Yang et al., 2021b; Zhang et al., 2023b; Peng et al., 2023; Yi et al., 2023). However, due to inter-modal disparities, a unified learning framework often results in imbalances of modalities. This phenomenon is defined as modality laziness, referring to situations where multi-modal performance is worse than single-modal performance (Du et al., 2023a). A common solution is to utilize late fusion to preserve uni-modal learning (Hessel & Lee, 2020; Yao & Mihalcea, 2022), or to employ knowledge distillation prior to modality fusion (Wang et al., 2020; Peng et al., 2022). In the ASR task, most efforts on multimodal fusion focus on audio-visual speech recognition (Afouras et al., 2018a; Hsu & Shi, 2022). However, although the modality laziness phenomenon is also reported in (Du et al., 2023a), limited research focuses on addressing it.

**Uncertainty Estimation in Auto-regressive Task.** The motivation of uncertainty estimation is to evaluate the reliability of a neural model's predictions, which is typically measured by Bayesian neural networks (BNNs) (Neal, 2012) and its varieties (Gal & Ghahramani, 2016; Han et al., 2022). (Malinin & Gales, 2020) first develop an ensemble-based uncertainty estimation framework for auto-regressive prediction. In the ASR task, uncertainty estimation is also explored for improving noise-robustness (Tran et al., 2014; Stouten et al., 2006), knowledge distillation (Kim et al., 2021), and intelligibility prediction (Tu et al., 2022). Furthermore, Zhang et al. (2023c) theoretically proves that uncertainty estimation can also be applied to modality fusion, where an energy score is utilized to determine a dynamic weight for each modality. This work extends the theory to the context of autoregressive decoding, which performs step-wise late fusion based on the uncertainty of LLMs' predictions.

## 2.1 A GENERATIVE FRAMEWORK OF ASR ERROR CORRECTION

Given the speech signal $X \in \mathbb{R}^l$, the ASR task aims to predict its textual transcription with $T$ sequential tokens $Y_T = (y_1, y_2, \cdots, y_T)$ with a neural network. HyPoradise dataset (Chen et al., 2023b) provides an informative N-best list consisting of $n$ hypotheses candidates $\hat{\mathcal{Y}}_n = \{\hat{Y}_1, \hat{Y}_2, \cdots, \hat{Y}_n\}$ using beam search, and then learn a hypotheses-to-transcription (H2T) mapping in a auto-regressive manner:

$$Y_T = \mathcal{M}_{H2T}(\hat{\mathcal{Y}}_n, \theta_l), \qquad P(Y_T|\hat{\mathcal{Y}}_n, \theta_l) = \prod_{t=0}^{T} P(y_t|Y_{<t}, \hat{\mathcal{Y}}_n, \theta_l) \qquad (1)$$

where $\theta_l$ denotes a pre-trained LLM with LoRA adapter, and $Y_{<t}$ denotes the history sequence $(y_1, y_2, \cdots, y_{t-1})$. It is worth noting that such a learning paradigm is text-only, as $X$ is not directly involved in the calculation of $P(Y_t)$. Consequently, it introduces extra expected data uncertainty when predicting transcription. In this work, we focus on integrating $X$ or its hidden representation into H2T mapping, which can be written as:

$$P(Y_T|\hat{\mathcal{Y}}_n, \theta_l, X) = \prod_{t=0}^{T} P(y_t|Y_{<t}, \hat{\mathcal{Y}}_n, \theta_l, X) \qquad (2)$$

More details about the HyPoradise dataset and relevant H2T learning are attached in Appendix A.1.

## 3 ACOUSTIC INFORMATION FUSION

In this part, we first illustrate different fusion strategies including early, mid, and late fusion. Then we concentrate on late fusion, and introduce the relevant techniques in the proposed UADF.

### 3.1 FUSION STRATEGY

Considering the long-range character of the raw speech signals $X$, we employ neutral network $\theta_a$ to extract its hidden representation. In this work, we investigate three common speech representations in a Transformer-based ASR model named: (i) $X_{tok}$: Speech tokens extracted by self-supervised learning, e.g., Wave2Vec (Schneider et al., 2019). (ii)$X_{enc}$: acoustic features in the last layer of the acoustic encoder, and (iii) $X_{dec}$: acoustic features in the last layer of the ASR decoder. These three kinds of representations range from shallow to deep, which corresponds to different fusion approaches shown in Figure 1.

***Early Fusion with Speech Tokens and Language Embedding***: An early fusion approach directly concatenates the speech tokens $X_{tok}$ and word embeddings $Y_{tok}$ before feeding into the first self-attention layer of LLM decoder, where $X_{tok}$ requires to be projected to the dimension of the $Y_{tok}$ to ensure compatibility. In practice, we follow the stacking approach introduced in (Fathullah et al., 2023) to reduce the length of $X_{tok}$. Subsequently, the $X_{tok}$ serves as prompt tokens that are fed into the LLM decoder, and perform auto-regressive decoding as follows:

$$P(Y_T) = \prod_{t=0}^{T} P(y_t|Concat(X_{tok}, Y_{<t}), \hat{\mathcal{Y}}_n, \theta_l) \tag{3}$$

Considering the $X_{tok}$ and $\hat{\mathcal{Y}}_n$ are from distinct modalities, prematurely fusing them may lead to modality laziness (Du et al., 2023b) due to modality gap, as LLMs can proficiently handle $\hat{\mathcal{Y}}_n$ while remains entirely unacquainted with $X_{tok}$.

***Mid-Fusion thorough Model Attention Merging:*** model attention merging is one recent neural adapter-based techniques (Lin et al., 2023; Radhakrishnan et al., 2023a) (i.e., Whispering-LLaMa) that utilizes the cross-attention mechanism in the LLM decoder to integrate $X_{enc}$ into the decoding process. Specifically, we utilize $X_{enc}$ as key $K(X_{enc})$ and value $V(X_{enc})$ matrices, then perform cross-attention using query $Q_{llm}$ in LLM decoder layer. The final layer output is obtained by summing $\hat{C}$ and original self-attention representation $C$ with a fixed weight $\lambda$:

$$\hat{C} = \text{softmax}(\frac{Q_{llm} \cdot (K(X_{enc}))^T}{\sqrt{d_k}}) \, V(X_{enc}) \qquad C = \text{softmax}(\frac{Q_{llm} \cdot (K_{llm})^T}{\sqrt{d_k}}) \, V_{llm}) \tag{4}$$

Two considerations should be addressed when applying mid fusion in LLMs: 1) the $X_{enc}$ requires to be aligned with $Q_{llm}$, as the latent dimensions are usually mismatched between the acoustic model and LLMs. A typical solution is adding a trainable adapter to ensure dimensional compatibility, which also serves as a modality converter from audio to text (Chen et al., 2023d; Yang et al., 2023). 2) Since mid fusion happens in each layer of LLMs, the tuning approach is expected to balance the perception of new modality with the retention of pre-trained knowledge.

In this paper, we employ a residual adapter with a down-up internal structure for modality transfer and dimension alignment. Furthermore, we keep most LLMs parameters frozen for the retention of pre-trained knowledge, and conduct both prompt-tuning and LoRA adapter (Yu et al., 2023) in each decoder layer to perceive acoustic representations $X_{enc}$. More details and discussion are attached in the Appendix A.3 and Appendix A.5.

***Late Fusion in Auto-regressive Decoding:*** A step-wise late-fusion happens in the auto-regressive decoding process that integrates decision-level information to predict the current token. Therefore, we decompose the $X_{dec}$ into $f_t^{asr}(X)$ according to step $t$, which is sequentially calculated by an independent encoder-decoder ASR model. Moreover, $f_t^{asr}(X)$ can be viewed as *logits* that indicates the probability distribution on the vocabulary space $V$. Although no feature-level alignment

is needed, late fusion requires a consistent decoding space between the ASR model and LLMs, and then performing weighting fusion to obtain $t$-step logits $f_t$:

$$f_t = w^{llm} f_t^{llm}(\hat{\mathcal{Y}}_n) + w^{asr} f_t^{asr}(X), \qquad \text{and } f_t^{llm}, f_t^{asr} \in \mathbb{R}^V. \qquad (5)$$

Specifically, considering the pre-existence of fusion techniques between ASR and LM, we herein elucidate the connection between our work and several existing approaches. Both mid fusion and deep fusion (Gulcehre et al., 2015) involve integration at the level of hidden features. However, mid fusion further capitalizes on the cross-attention mechanism of the LLM decoder. Late fusion and shallow fusion (Kannan et al., 2018) exhibit similarities, yet LLM in late fusion scheme can uniquely leverage the GER to independently predict transcriptions without the assistant of ASR model. Additionally, unlike cold fusion (Sriram et al., 2017), late fusion avoids the subsequent training process after fusing the information in auto-regressive decoding.

Considering late fusion fuses information in the decision-level stage, it maximally mitigates the emergence of modality laziness problems. The upcoming two chapters will be dedicated to the exploration of two key challenges in late fusion: (i) *how to calibrate the token-level logits $f_t$*, and (ii) *how to determine the fusion weights $w^{llm}$ and $w^{asr}$*.

## 3.2 CALIBRATION

The significance of the calibration in late fusion arises from the *over-confidence* phenomenon (Bai et al., 2021) in neural networks, which indicates the confidence score of models is usually higher than their accuracy. In the ASR task, the training process utilizes the Teacher-forcing technique that the $Y_{<t}$ in E.q.( 2) are drawn from the ground-truth sequence. However, the trained model has to rely on its own prediction during inference as ground truth is not available. This mismatch leads to exposure bias that exacerbates the over-confidence when calculating the probability of the current token $P_{y_t}$ during auto-regressive decoding. Furthermore, prior study (Mukhoti et al., 2020) demonstrates that over-confidence can seriously hurt the ensemble performance when integrating the logits of classification models. To alleviate this issue, we adopt a temperature scaling approach introduced in (Kumar et al., 2022). Specifically, we establish two temperatures $\tau$ to match up the confidences of models with their average accuracies on a small validation set:

$$\text{Conf}(f, \tau) = \frac{1}{n_{dec}} \sum_{i=1}^{n_{dec}} \max \text{softmax}(\frac{f_i}{\tau}) \qquad (6)$$

where $n_{dec}$ denotes all decoding steps accumulated from the samples in the validation set. When $\tau \to 0$ we obtain a uniform distribution, and when $\tau \to \infty$ we obtain a Dirac distribution on the most likely output. In practice, we determine the $\tau_1$ for LLMs and $\tau_2$ for ASR using a binary search algorithm based on token error rate (TER):

$$\text{Conf}_{llm}(f^{llm}, \tau_1) \approx 1 - \text{TER}_{llm}(f^{llm}), \quad \text{Conf}_{asr}(f^{asr}, \tau_2) \approx 1 - \text{TER}_{asr}(f^{asr}) \qquad (7)$$

There are several alternative approaches for TER calculation, as the divergence between ASR and LLMs inevitably leads to different history sequences $Y_{<t}$. To unify it, we update the $Y_{<t}$ using greedy strategy based on the calibrated probability $f_t$ for token $y_t$, which is written as:

$$f_t = w^{llm} \text{softmax}(\frac{f_t^{llm}}{\tau_1}) + w^{asr} \text{softmax}(\frac{f_t^{asr}}{\tau_2}) \qquad (8)$$

## 3.3 UNCERTAINTY-AWARE DYNAMIC FUSION

An intuitive method to measure $w^{llm}$ and $w^{asr}$ is to estimate two constants according to WER performance on the validation set. However, such a static fusion strategy leads to a higher upper bound of generalization error compared with dynamic fusion, which has been theoretically proven using Rademacher complexity (Bartlett & Mendelson, 2002) in multi-modal classification (Zhang et al., 2023c). More importantly, Zhang et al. (2023c) theoretically identifies the connection between dynamic multimodal fusion and uncertainty estimation. This connection is relevant to our motivation: we focus on fusing acoustic information to tackle the data uncertainty in the H2T learning of LLMs. Typically, the uncertainty of a $y_t$ by LLMs is given by the entropy of the predictive posterior:

$$\mathcal{U}_t^{llm} = -P(y_t) \cdot \log P(y_t), \qquad \text{where } P(y_t) = \text{softmax}(\frac{f_t^{llm}}{\tau_1}) \qquad (9)$$

A large $\mathcal{U}_t^{llm}$ means a large uncertainty when LLMs predict the current token, which requires more acoustic compensation from $f_t^{asr}$. Since late fusion incorporates only two modalities, and considering the predominant role of LLMs within them, we set the $w_t^{llm}$ as 1 and dynamically modulate $w_t^{asr}$ in terms of $\mathcal{U}_t^{llm}$. Therefore, the uncertainty-aware dynamic fusion in auto-regressive decoding can be written as:

$$P(Y_T) = \prod_{t=0}^{T} \mathrm{softmax}(\mathrm{softmax}(\frac{f_t^{llm}}{\tau_1}) + (\mathrm{sigmoid}(\mathcal{U}_t^{llm}) - \beta)\,\mathrm{softmax}(\frac{f_t^{asr}}{\tau_2})) \quad (10)$$

where $\beta$ is a hyper-parameter with a default value of 0.5. From E.q. 10 we observe that if the LLM is extremely confident after calibration ($\mathcal{U}_t^{llm} \to 0^+$), then the final decision could completely rely on its own decision ($\mathrm{sigmoid}(\mathcal{U}_t^{llm}) - 0.5 \to 0^+$). Otherwise, the weight of the ASR model increases with the increase of $\mathcal{U}_t^{llm}$.

## 4 EXPERIMENT

### 4.1 DATASET

**HyPoradise**[1] (Chen et al., 2023b) is a generative error correction benchmark for LLM-enhanced ASR task, which contains more than 316K hypotheses-transcription pairs collected from mainstream ASR corpus. Specifically, each utterance is equipped with at least 5 hypotheses that are transcribed by a Whisper-large-v2 model with beam search decoding. In this work, we employ WSJ and ATIS as clean condition and CHiME-4 as noisy condition from HyPoradise, and more statistic details are in Appendix A.1 and Table 4. In this paper, we select the WSJ (Paul & Baker, 1992; Garofalo et al., 2007), ATIS (Hemphill et al., 1990), CHiME (Vincent et al., 2016), and LRS3 (Afouras et al., 2018b) datasets to evaluation the proposed methods. More details can be found in Appendix A.2.

### 4.2 SETUP

**H2T Learning.** We employ LLaMA-7B[2] from Huggingface as the foundation model for H2T learning. A low-rank adapter is inserted into each layer of LLaMA with the rank of 8. We use a uniform prompt template to transform the N-best list into inputs suitable for LLMs. More training details and hyper-parameters can be found in Appendix A.1. Additionally, we employ the *GER* method as a baseline and report the results in the next section.

**Early and Mid Fusion baselines.** $X_{tok}$ is extracted from raw speech signal by Wav2vec2-large and HuBERT pre-trained models. They have both been trained by CTC loss on the LibriSpeech dataset. Since the dimension of $X_{tok}$ is 1024, we stack 4 $X_{tok}$ to align with the dimension of LLaMA's word embedding and reduce the length of $X_{tok}$. $X_{enc}$ is extracted by Whisper encoder.

**Uncertainty Aware Dynamic Fusion (UADF).** We employ a Whisper-*tiny* model to provide $X_{dec}$ for late fusion. Since it can independently calculate WER, we term it as ASR-*only* baseline in the experiments. To align the decoding space with LLMs, we fix the encoder and finetune the decoder using LLaMA's word embeddings, Tokenizer, and special tokens. We randomly select a small validation set with 200 training examples from the training set to determine $\tau_1$ and $\tau_2$ using binary search, as well as selecting the best model. With the same validation set, we also utilize the grid search to find out the best-fixed weight for LLMs and ASR models and perform static late fusion as our baseline. Additionally, the $\beta$ in E.q. 10 is set as default 0.5 for both ATIS and WSJ.

## 5 RESULT AND ANALYSIS

In this section, we conduct experiments and answer the following questions: (i) What is the performance of different fusion strategies when integrating acoustic information into LLMs, (ii) Does the proposed UADF method surpass its counterparts (static fusion) using late fusion, and (iii) How does the generalization ability of UADF, and can it be seamlessly applied to other ASR-related tasks?

---

[1]https://huggingface.co/datasets/PeacefulData/HyPoradise-v0
[2]https://huggingface.co/decapoda-research/llama-7b-hf

Table 1: WER (%) and WERR results of early, mid, and late fusion on ATIS and WSJ dataset. "*W2v.*", "*Hub.*" and "*Whis.*" indicate Wav2vec2-large, HuBERT and Whisper model, respectively. "*Conc.*", "*Atten.*", and "*Stat.*" indicate concatenation, cross-attention and static fusion strategies introduced in 3. "GER" denotes the H2T results of LLM that is consistent across the three fusion methods.

| Acoustic Info. | Fusion | | GER | | ASR-*only* | | WER ↓ | | WERR ↑ | |
|---|---|---|---|---|---|---|---|---|---|---|
| | *where* | *how* | *ATIS* | *WSJ* | *ATIS* | *WSJ* | *ATIS* | *WSJ* | *ATIS* | *WSJ* |
| $X_{tok}$ by W2v. | *early* | Conc. | 1.61 | 2.83 | - | - | 2.16 | 3.21 | -34.2% | -13.4% |
| by Hub. | | | | | - | - | 2.02 | 3.11 | -25.5% | -9.9% |
| $X_{enc}$ by Whis. | *mid* | Atten. | 1.61 | 2.83 | - | - | 1.75 | 2.59 | -8.7% | 8.5% |
| $X_{dec}$ by ASR | *late* | *Stat.* | 1.61 | 2.83 | 4.67 | 9.21 | 1.36 | 2.55 | 15.5% | 9.9% |
| | | UADF | | | | | **1.24** | **2.47** | **23.0%** | **12.7%** |

Table 2: Ablation study of WER (%) and WERR results on the ATIS dataset based on UADF using late fusion. The difference between the system ID-1 to ID-3 is the different performance of ASR-*only* model ($X_{dec}$), and the system ID-4 to ID-5 varies based on the different combination of "*Cali.*" and "*Dyn.*". "Static" does not utilize either "*Cali.*" and "*Dyn.*".

| System ID | GER | ASR-*only* ($X_{dec}$) | ID-C | | Static | | UADF | | | |
|---|---|---|---|---|---|---|---|---|---|---|
| | | | WER | WERR | WER | WERR | Cali. | Dyn. | WER | WERR |
| 1 | | 12.16 | 2.41 | -49.7% | 1.51 | 6.2% | ✓ | ✓ | 1.52 | 5.6% |
| 2 | 1.61 | 8.22 | 1.96 | -21.7% | 1.45 | 9.9% | ✓ | ✓ | 1.39 | 13.7% |
| 3 | | 4.67 | 1.57 | 2.5% | 1.36 | 15.5% | ✓ | ✓ | **1.24** | **23.0%** |
| 4 | 1.61 | 4.67 | 1.57 | 2.5% | 1.36 | 15.5% | ✓ | ✗ | 1.33 | 17.4% |
| 5 | | | | | | | ✗ | ✓ | 1.39 | 13.7% |

We employ the word error rate (WER) and word error rate reduction (WERR) to evaluate the performance. A lower WER parameter signifies better performance, while a higher WERR indicates a greater improvement relative to GER.

## 5.1 EFFECT OF FUSION STRATEGIES

We first report the WER performance on ATIS and WSJ for different fusion strategies in Table 1. ASR-*only* is calculated by $X_{dec}$ and thus only appears in late fusion. From Table 1, we observe that: (i) *Early fusion* performs slightly below the GER baseline in terms of WER, regardless of using Wav2vec2-large or HuBERT as a tokenizer. The underlying reason is intuitive: when we concatenate $X_{token}$ and $\hat{\mathcal{Y}}_n$, the language model has no knowledge of $X_{token}$ but is well-acquainted with the $\hat{\mathcal{Y}}_n$, leading to the occurrence of modal laziness. In other words, the acoustic information $X_{token}$ would be regarded as a form of linguistic "noise" if we treat the concatenated $X_{token}$ and the n-best list as prefix tokens. (ii) *Mid fusion* on the WSJ dataset shows better performance than WSJ due to the larger data amount. It indicates that mid fusion requires more training examples to overcome the modal disparities in cross-attention. (iii) *Late fusion* achieves considerable performance gains compared with GER baseline, where UADF respectively reduces the relative WER by 23.0% and 12.7% on ATIS and WSJ datasets. Surprisingly, despite the ordinary performance of the ASR model, a static weight sum approach yields better WER results than GER. Additionally, we observe that when the ratios of $w^{llm}$ and $w^{asr}$ fall within a certain range (e.g., $w^{llm}$ / $w^{asr}$ = $4 \pm 2$ on ATIS), the combination produces highly similar results.

To visualize the effect of UADF, we conducted a case study to show how the UADF performs late fusion to correct LLM's token-level decision. Figure 2 is a real case drawing from the decoding process of ATIS test set. In this case, LLM predicts the current token as ID-5521 ("how") but with high uncertainty ($\mathcal{U}_t^{llm}$ = 9.91). According to E.q 10, UADF allocates a high weight (≈ 0.5) to the ASR model, resulting in a token ("all") with ID-484 as the final decision that is consistent with ground truth token.

Table 3: WER (%) and WERR results of Noise robust ASR results on Chime-4 dataset. The best results are in bold.

| Noise Type | ASR-*only* | GER | Static | | UADF | |
|---|---|---|---|---|---|---|
| | | | *WER* | *WERR* | *WER* | *WERR* |
| *bus* | 12.45 | 8.67 | 8.05 | 7.2% | **7.98** | **8.0%** |
| *caf* | 11.48 | 6.96 | 6.37 | 8.5% | **6.22** | **10.6%** |
| *ped* | 11.36 | 5.49 | 4.96 | 9.8% | **4.82** | **12.2%** |
| *str* | 12.28 | 5.86 | 5.28 | 9.9% | 5.28 | 9.9% |
| *Avg.* | 11.89 | 6.75 | 6.17 | 8.6% | **6.08** | **9.9%** |

## 5.2 EFFECT OF UADF

We then conduct an ablation study to analyze the effectiveness of UADF in late fusion. There are three primary factors that influence the performance of UADF, as shown in Table 2, which are: the performance of ASR-*only* model ($X_{dec}$), the calibration operation ("*Cali.*"), and the uncertainty-based dynamic fusion ("*Dyn.*"). "ID-C" is a late fusion-based baseline that only utilizes the same calibration method with two same constants $w^{llm}$ and $w^{asr}$ for decision in classification (Kumar et al., 2022). "Static" is to estimate the value of $w^{llm}$ and $w^{asr}$ according to WER on the validation set without calibration.

In Table 2, we observe that: (i) As the ASR-*only* models (ID-1 to ID-3) exhibit progressively better performance, the late fusion results become better in terms of WER. It is noteworthy that even a modest-performing ASR-*only* model (12.16%) with UADF can yield substantial performance gains (1.61% → 1.52%) compared with GER. (ii) From the performance of "ID-C", it is evident that performing calibration in isolation without weight searching does not enhance GER performance. This is primarily due to the complexity of auto-regressive decoding. However, calibration plays a crucial role in UADF, since it can mitigate the issue of overconfidence in our models, thereby encouraging diversity in fusion decisions. (iii) UADF can achieve better WER performance than static fusion, as it adaptively assimilates the decision of $X_{dec}$ in terms of uncertainty. Furthermore, compared with static fusion, UADF avoids searching the fusion weight on a validation set.

To support viewpoint (ii) and illustrate the importance of calibration in UADF, we visualize the accuracy, confidence, and token distribution on the ATIS test set in Figure 3, where the histogram denotes the token distribution according to LLM's confidence, and the "×" denotes the actual average accuracy based on each confidence interval. In the left part, LLM shows the obvious over-confident phenomenon: more than 98.9% of token predictions fall within the confidence interval of 0.9 to 1.0, and their average confidence is 99.9%, which is higher than the true accuracy of 96.5%. In other intervals, the confidence is also higher than the actual accuracy, since all "×" are below the dashed line. After calibration, the overconfidence issue is significantly alleviated, as shown in the right part of Figure 3. 93.7% tokens fall within the confidence interval of 0.9 to 1.0, while the average confidence has dropped to 97.03%, which is similar to the accuracy of 96.5%. Additionally, calibration can affect subsequent uncertainty estimation, enabling the identification of token decisions where the LLM performs poorly (e.g., the case in Figure 2), and facilitating the dynamic incorporation of decision information from the ASR model.

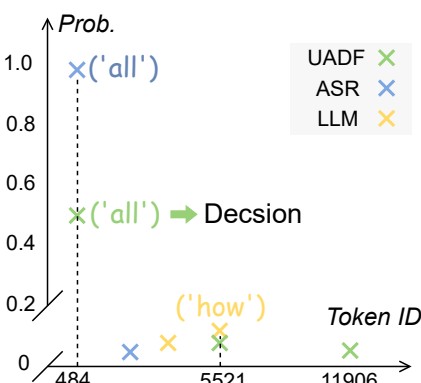

Figure 2: Case study on a high uncertainty ($\mathcal{U}_t^{llm}$ is 9.91) example. Top-2 candidates from LLM are displayed while the "how" is a wrong prediction. UADF corrects the results to "all" according to the decision of the ASR model.

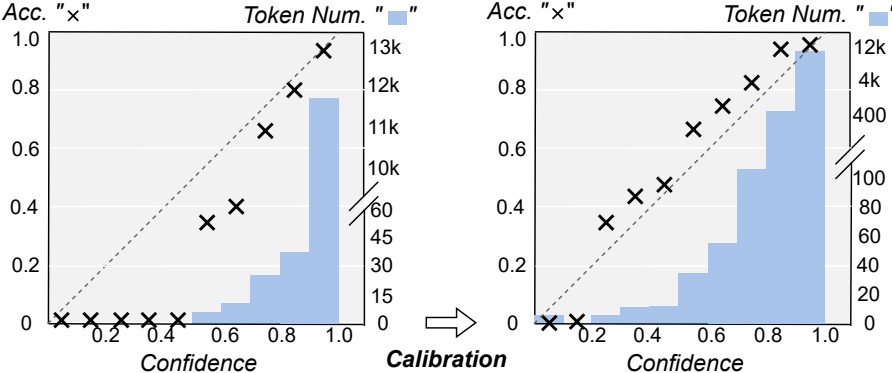

Figure 3: The visualization of before (left) and after (right) calibration for LLM in UADF. The dashed line represents the ideal relationship where confidence and accuracy are perfectly matched. The blue bar indicates the token distribution under different LLM's confidence intervals, and "×" indicates the actual average accuracy based on each confidence interval.

## 5.3 GENERALIZATION OF UADF

We first consider examining the UADF's generalization on noise-robust ASR task, as background noise can increase the variability among hypotheses in the N-best list, thus leading to higher uncertainty for LLMs when predicting transcription. The results of UADF on the CHiME-4 dataset are reported in Table 3 in terms of noise categories. We observe that the performance of ASR-*only* GER slightly drops due to noise interference compared with ATIS and WSJ. However, UADF approach can yield significant performance gains across different noise environments. Furthermore, as same in clean conditions, UADF outperforms the static baseline due to the sigmoid function, which can normalize the high uncertainty of individual tokens, mitigating over-reliance on the ASR model.

We then validate the effect of UADF in the audio-visual speech recognition (AVSR) task, where noise-invariant visual modality is utilized to provide compensation information for speech recognition. It is worth noting that the modality laziness phenomenon is particularly in AVSR because the system tends to overly rely on the audio modality due to its higher recognition ease. More introduction and discussion about AVSR are attached in Appendix A.4. With the proposed UADF, we perform late fusion on AV-HuBERT baseline (Shi et al., 2022b) and a pre-trained lip-reading model (Shi et al., 2022a). Besides static fusion, we employ MSRL (Chen et al., 2023c) as a baseline, which integrates two models in a reinforcement learning-based manner. In Table 6, we observe that multimodal AV-HuBERT achieves worse performance than unimodal V-HuBERT due to modality laziness. Accordingly, all three methods can effectively improve noise-robustness by reusing the independent visual modality. Furthermore, UADF surpasses static fusion in all conditions in terms of WER and achieves comparable performance with MSRL. Notably, MSRL requires an extra training process for reinforcement learning while our UADF is training-free.

## 6 CONCLUSION

In this paper, we ask a basic yet well-discovered question: how can audio information be integrated into Large Language Models (LLMs) for GER-based speech recognition tasks? After exploring multiple fusion strategies at different levels, we present a simple yet effective solution UADF that performs late fusion in the auto-regressive decoding process. Benefiting from uncertainty estimation of LLM outputs, UADF dynamically assimilates information from the audio modality, leading to more reasonable token-level decisions. Experimental evidence demonstrates that our method can avoid modality laziness, yielding better WER performance gain to the GER compared with other fusion strategies. Additionally, UADF seamlessly adapts to noise-robust ASR as well as AVSR.

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

# A  APPENDIX

## A.1  HYPORADISE AND H2T LEARNING

**Hyporadise** dataset provides more than 316k hypotheses-transcription pairs, and Table 4 shows the statistic information of WSJ and ATIS we used in this paper. Each utterance is equipped with at least 5 hypotheses that are transcribed by a Whisper-large-v2[3] model with beam search decoding.

Table 4: Statistics in terms of the number of hypotheses-transcription pairs and average utterance length on WSJ and ATIS.

| Source | Domain Category | Training Set | # Pairs | Length | Test Set | # Pairs | Length |
|--------|-----------------|--------------|---------|--------|----------|---------|--------|
| WSJ | Business news | *train-si284* | 37,514 | 17.5 | *dev93* | 503 | 16.7 |
|     |               |               |        |        | *eval92* | 333 | 17.3 |
| ATIS | Airline info. | *train* | 3,964 | 12.4 | *test* | 809 | 11.3 |
| CHiME4 | Noise | *train* | 8,738 | 17.0 | *test-real* | 1,320 | 16.4 |

**H2T learning** indicates learning the mapping relationship from the N-best hypotheses list to transcription, which requires the linguistic information of LLMs. The motivation behind this method is to harness the token-level information present within the N-best list, while mainstream ASR methods usually only output the first hypothesis and discard others. Take a Confermer-based ASR model trained on LibriSpeech[4] (WER 1.8) as an example, the second hypothesis has a 14% probability of having a lower WER than the first hypothesis. Furthermore, given a wrong token in the first utterance, there is a 34% probability of finding the correct token in the second utterance. Furthermore, if we have an oracle re-ranking method to choose the best hypothesis, the WER would be 1.0. If we have an oracle compositional method that uses token-level information to predict transcription, the WER would be 0.6. Both 1.0 and 0.6 surpasses state-of-the-art performance by a large margin. To perform H2T learning, we utilize an instruction-following prompt template shown as follows:

*"Below is a best-hypotheses that is transcribed from an automatic speech recognition system. Write a response to predict the true transcription using the tokens from other-hypotheses.### best-hypothesis:$\{1^{st}$ utterance$\}$### other-hypothesis:$\{2^{nd} \sim 5^{th}$ utterances$\}$ ###Response:"*

We employ a LLaMA as a foundation model from huggingface, the learning rate is set as $1e^{-4}$, and the batch size is 128. For the low-rank adapter, we implement by peft [5], where the rank configuration of rank $r$ is set as 8. For early fusion, Wav2vec2-large[6] and HuBERT[7] pre-trained models are used to extract $X_{tok}$ from raw speech signals, which have been trained by CTC loss on the LibriSpeech dataset.

To show the effect of H2T learning, we report our reproduced GER results with LLaMA-7b in Table 5. 1-*best* denotes the WER of the first hypothesis with highest probability in the given N-best list. Notably, this probability is provided by the ASR beam search decoding. The $o_{nb}$: WER of the "best hypothesis" in N-best hypotheses list, which can be viewed as the upper bound performance of any reranking based methods. The compositional oracle method $o_{cp}$ is the achievable WER using "all tokens" in N-best hypotheses list. We also attach the results of a rerank baseline "$LM_{rank}$" and proposed UADF for comparison.

## A.2  INTRODUCTION OF DATASETS

**WSJ** (Wall Street Journal) (Paul & Baker, 1992; Garofalo et al., 2007) is a widely-used ASR corpus that focuses on the domains of business news and financial data. The training set includes 37514

---

[3] https://huggingface.co/openai/whisper-large-v2
[4] https://www.openslr.org/12
[5] https://github.com/huggingface/peft
[6] https://huggingface.co/facebook/wav2vec2-large
[7] https://huggingface.co/facebook/hubert-large-ll60k

Table 5: WER (%) results of GER and UADF with more baselines. "*" denoted the WER is reproduced in this paper.

| Dataset | 1-*best* | $LM_{rank}$ | GER | UADF | Oracle | |
| | | | | | $o_{nb}$ | $o_{cp}$ |
|---|---|---|---|---|---|---|
| ATIS | 8.9 | 6.9 | 1.61* | 1.29* | 5.2 | 1.1 |
| WSJ | 4.5 | 4.3 | 2.83* | 2.47* | 4.1 | 1.2 |
| CHiME | 11.1 | 11.0 | 6.75* | 6.08* | 9.1 | 2.8 |

utterances from 101 speakers in a clean environment. The test set consists of *dev93* (with 503 utterances) and *eval92* (with 333 utterances) and we report the average WER results on them.

**ATIS** (Airline Travel Information System) (Hemphill et al., 1990) is an ASR dataset that concentrates on the domain of air travel information, such as flight times, prices, and availability. It contains a training set with 3964 utterances and a test set with 809 utterances that are collected from more than 500 different speakers.

**ChiME-4** (Vincent et al., 2016) is a dataset that is widely used in noise-robust ASR task. It includes real and simulated noisy recordings in four noisy environments, i.e., bus, cafe, pedestrian area, and street junction. This work employ the test-*real* as test set that is recorded in real noisy conditions.

**LRS3** (Lip Reading Sentences) (Afouras et al., 2018b) is the largest public multimodal dataset for audio-visual speech recognition (AVSR) tasks. It contains more than 400 hours of speech data with paired face images of speakers. We utilize LRS3 to demonstrate the generalization ability of UADF on AVSR tasks.

### A.3 DETAILS OF MID FUSION

To build cross-modal fusion during the intermediate transformer layers, we introduce a neural adapter based mid-fusion, which inspired by multi-modal attention merging (Sung et al., 2023; Hung et al., 2023; Radhakrishnan et al., 2023a). To fine-tune fused models, we integrate two residual adapter modules (Houlsby et al., 2019; Radhakrishnan et al., 2023b; Chen et al., 2023e) ($A_L^i$ and $A_W^i$) subsequent to the self-attention modules ($SA_F^i$) of the stabilized LLaMA model within each layer. The adapter $A_L^i$ is the module in layer $i$ assigned to refine the LLaMA model, employing a scaled dot product attention mechanism. Conversely, the adapter $A_W^i$ is used in layer $i$ for the amalgamation of pre-trained Whisper features with the LLaMA model, adhering to a wise-layer fused decoder approach. We employ the Adam optimizer and conduct experiments with learning rates of $1e^{-2}$, $1e^{-3}$, and $5e^{-4}$, selecting the best development losses over 25 epochs for evaluation. We select a batch size of 32 and apply a weight decay of $1e^{-2}$. For the residual adapter's setup, we fix bottleneck dimensions of 32 after ablation studies. A similar design in vision and acoustic-based mid-fusion can also be referred to (Lin et al., 2023), where this baseline can be considered as its language and acoustic-based variant.

### A.4 DISCUSSION OF AUDIO-VISUAL SPEECH RECOGNITION

Audio-visual speech recognition (AVSR) represents a cutting-edge interdisciplinary task that integrates principles from both auditory and visual processing domains to enhance speech recognition capabilities. This task involves the synchronous analysis of audio signals and visual cues, particularly lip movements and facial expressions, to accurately decipher spoken language. AVSR systems leverage the complementary nature of audio and visual information to improve recognition accuracy, especially in noisy environments where traditional audio-only systems might struggle. This technology not only holds promise for advancing human-computer interaction but also offers significant improvements in accessibility for individuals with hearing impairments. Mainstream AVSR methods focus on learning modality-invariant representations by integrating audio and visual modalities into a common subspace. Typically, they employ a separated encoder for speech and image input, and then concatenate the hidden representation after alignment (Afouras et al., 2018a; Chen et al., 2023f). This learning pattern can easily lead to modal laziness because the audio modality is much

Table 6: WER (%) and WERR results of AVSR task on the LRS-3 by late fusion. "Babble" is the noise drawn from (Snyder et al., 2015). "*" indicates the need for an additional training procedure.

| SNR (Babble) | AV-HuBERT audio-visual | V-HuBERT visual-only | Static w | | MSRL* | | UADF | |
|---|---|---|---|---|---|---|---|---|
| | | | WER | WERR | WER | WERR | WER | WERR |
| -10 | 30.3 | | 23.4 | 22.8% | 22.3 | 26.4% | **21.8** | **28.1%** |
| -5 | 13.5 | | 11.4 | 15.6% | 11.3 | 16.3% | **10.7** | **20.7%** |
| 0 | 4.9 | + 26.9 | 4.8 | 2.0% | **4.5** | **8.2%** | 4.6 | 6.1% |
| 5 | 2.5 | | 2.8 | -12.0% | **2.3** | **8.0%** | 2.6 | -4.0% |
| Avg. | 12.8 | | 10.6 | 17.2% | 10.1 | 21.1% | **9.9** | **22.7%** |
| Clean | 1.45 | 26.9 | 1.42 | 2.1% | **1.33** | **8.3%** | 1.36 | 6.2% |

easier to recognize than visual information, causing neural networks to gradually ignore the role of the visual modality.

Recently, Shi et al. (2022a) proposes a self-supervised learning approach to learn better modality-invariant representation for AVSR task. With large amount of pre-training data, the AV-HuBERT achieves remarkable performance on LRS-3 dataset. However, based on AV-HuBERT, Chen et al. (2023c) still report the modality laziness problem for modality-invariant representation as shown in Figure 4: the modality-invariant representation (green line) exhibits a susceptibility to noise interference. More importantly, when SNR is smaller than a threshold $\alpha$, the multimodal representation perform worse than visual-only representation. To address it, Chen et al. (2023c) proposes a reinforcement learning based-method to reuse the visual modality representation in auto-regressive decoding process. It construct a trainable policy network to predict the final token probability distribution. We add MSRL for

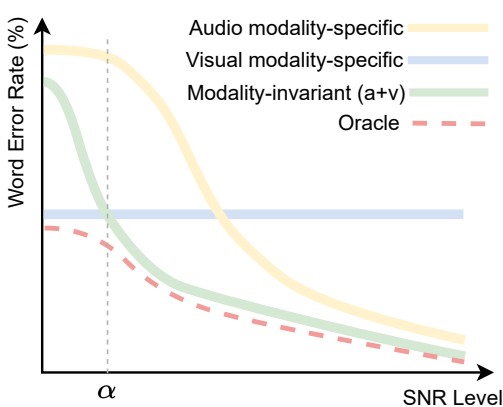

Figure 4: Modality laziness reported in (Chen et al., 2023c), where SNR level denotes the quality of speech.

comparison results in Table 6. For static fusion baseline, the $w$ of AV-HuBERT is {0.5, 0.65, 0.7, 0.75, 0.85} for SNR {-10, -5, 0, 5, clean}, and the wight of AV-HuBERT is $1 - w$. For UADF implementation, we observe though WER is high, the AV-HuBERT still exhibit over-confidence tendency that is higher than actual accuracy. We replace LLM using AV-HuBERT and estimate the uncertainty after calibration, and the $\beta$ is set as {0, 0.4, 0.5, 0.5, 0.5} respectively.

### A.5 DISCUSSION OF N-BEST LIST

For a fair comparison, we involve the N-best list in early fusion and mid fusion. However, this operation may introduce a potential issue in early fusion, where the LLM learns that it can predict the answer solely based on the n-best list, thus overlooking the acoustic tokens $X_{tok}$. To this end, we try to remove the n-best list and only employ $X_{tok}$ as prefix tokens in decoding. However, we found that the LLM is unable to identify $X_{tok}$ through low-rank tuning, and this might be attributed to the limited amount of training data. We leave it as our future work.

