# OpenReview forum: "It's Never Too Late: Fusing Acoustic Information into Large Language Models for Automatic Speech Recognition"
_ICLR.cc/2024/Conference — ICLR 2024 poster_

### Official Review · Reviewer_GWCM · 2023-10-29

**Soundness:** 3 good
**Presentation:** 3 good
**Contribution:** 4 excellent
**Rating:** 10
**Confidence:** 4

**Summary:**

To integrate acoustic information into the speech recognition output error correction using large language models (LLMs), the authors of this paper compare three fusion methods: early, mid, and late fusions. To further improve late fusion, this paper investigates adding uncertainty and proposes a new approach, Uncertainty-Aware Dynamic Fusion (UADF), introduced to integrate acoustic information, significantly enhancing word error rate (WER) and showing potential in audio-visual speech recognition.

**Strengths:**

Overall, this paper's structure is easy to follow.
It reminds me of previous milestone works about RNN decoders: Shallow fusion, Cold fusion, and Deep fusion. And this paper also has the potential to be a milestone.

The UADF is novel in this paper, and the improvements are guaranteed shown in Tables 1 and 2.

**Weaknesses:**

ICLR conference is less specialized for speech researchers than ICASSP and INTERSPEECH. A brief and precise background introduction to the speech recognition framework is needed.

For the three frameworks in Figure 1, it needs to be clarified: where is N-best from? The N-best and the X_tok, X_enc, and X_dec are from different types of models according to the descriptions. It is better to mention these differences in the figures.

"language models have been widely utilized in ASR tasks over the past two decades," maybe there is a better way to express it; some earlier efforts are ignored.

Additionally, I believe you want to demonstrate that the proposed method is also applicable to audio-visual tasks. This requires a more detailed description; merely using a single paragraph here is insufficient.

**Questions:**

The N-best are from WavLM and Whisper in GER-based H2T neurlPS2023. This paper is actually implementing a system combination, which, of course, will bring accuracy improvement. So, this work can be extended to more wider tasks.

---

> ### Author Response · Authors · 2023-11-15
> **Response for Reviewer GWCM**
>
> We sincerely appreciate Reviewer 1EWw for considering that this work is novel and easy to follow. Now we respond to your concerns and questions:
>
> - **Q1: More background for ASR and AVSR**
>
>     Thanks for your constructive suggestion. We have modified the related work Section for a more thorough literature review of ASR
>     technique, and more earlier efforts are reviewed. Meanwhile, we add more introduction to AVSR in Appendix A.4.
>
> - **Q2: Where is N-best from?**
>
>     We have modified the caption of Figure 1 to introduce the source of N-best list.
>
>
> - **Q3: Extending the proposed method to wider tasks**
>
>     Thanks for your suggestion. The UADF is potential to be applied in multimodal task with a auto-regressive decoding process, e.g.,
>     image captioning, where an image-to-text model is designed to describe the input image with natural language. Since the
>     mainstream image caption models are auto-regressive decoding, we argue that an external language model can be dynamically
>     fused to improve the quality of predicted captions.  We leave it as our future work.
>
> Thanks again for your time and patience.

---

> > ### Comment · Reviewer_GWCM · 2023-11-15
> > **good paper**
> >
> > Many thanks for your reply. I have no other questions.

---

### Official Review · Reviewer_3ovr · 2023-10-29

**Soundness:** 3 good
**Presentation:** 3 good
**Contribution:** 3 good
**Rating:** 6
**Confidence:** 5

**Summary:**

This paper builds upon the recently proposed “generative error correction” (GEC) paradigm for speech recognition (ASR). The previous GEC work used a text-only LLM to map from a list of ASR hypotheses to a single transcription. In this paper, the authors propose to add acoustic information to this mapping process by including speech representation in the GEC. They experiment with several such “fusion” strategies (so-called early, mid, and late fusion), and finally show that late fusion with uncertainty awareness works best. Evaluations are conducted on a number of ASR benchmarks, showing significant WER improvements compared to GEC or ASR-only baselines, and the method is also shown to generalize to audio-visual ASR.

**Strengths:**

1. Using LLMs for ASR error correction is an interesting idea and has shown strong results in the past. ASR models often perform badly on rare words such as named entities, and LLMs can be a useful component to improve recognition of such words. In the earlier GEC work, only the top hypotheses of the ASR model was used for the final transcription. Since these hypotheses may not always contain information from the correct word, it is natural to also use the ASR information more directly. With these considerations, the ideas in this paper are relevant and a logical continuation in this line of work.

2. Incorporating modality-specific uncertainty using the UADF approach is a novel and interesting technique, and seems to provide reasonable improvements compared to static weighting of modalities. For example, in Table 4, we see that particularly on very low SNR conditions, there is a large improvement in WER when using UADF instead of static weights. I also appreciate that the authors show ablation results and analysis for the UADF strategy in Section 5.2.

3. In addition to traditional ASR benchmarks, the authors show that their UADF strategy can be used for audio-visual ASR, where it gives a small improvement over AV-HuBERT.

**Weaknesses:**

My main concerns about the paper are about the presentation (in terms of relation with prior work) and comparison with baselines in the experiments, which I will detail below.

### This model in the context of prior work

In order to make this point clear, let me briefly summarize some key events in the history of the use of language models (LMs) in ASR.
- ASR was formulated as a noisy channel model using the Bayes rule P(W|X) = P(X|W)P(W). The two distributions were named an acoustic model and a language model, respectively. The LM was trained separately on source text, and only used for decoding [1].
WFST-based decoding provided a way to create a incorporate n-gram LMs into the decoding graph for efficient **first-pass decoding** [2].
- Incorporating larger n-gram LMs was hard (since the decoding graph explodes), so researchers used them instead in **second-pass rescoring** in both offline and on-the-fly settings [3, 4], by subtracting the original LM scores and adding back the larger LM’s scores.
- The Bayesian formulation still made sense in the era of hybrid HMM-DNN model, even when the acoustic models were discriminatively trained, since the scores could be interpreted as pseudo-likelihoods by subtracting an appropriate prior, and so the same decoding/rescoring framework carried over.
- In the era of “end-to-end” ASR, the models are trained to directly estimate P(W|X), and so log-linear interpolation with an external LM is not interpretable in the Bayesian sense anymore. Yet, practitioners still do this and call the process “fusion” (e.g. shallow fusion, cold fusion, etc. [5, 6]) — this is analogous to using LMs in the first-pass decoding.
- Analogous to the second-pass rescoring, we can obtain candidates using beam search on an ASR model, and then re-rank them with an externally trained LM (e.g. as done in the original LAS paper). So far, this “rescoring” step only manipulates the hypotheses, without considering the acoustic information. Let’s call this model (A).
- More recently, a two-pass E2E ASR model was proposed which has an encoder shared between a streaming RNN-T model and a full-context LAS decoder [7]. The idea was to perform a first-pass decoding with the RNN-T head, and use the decoded hypotheses along with the encoder representations to rescore using the LAS head in a *acoustic-guided rescoring* technique.
- In [8], the authors further built upon this two-pass model by proposing a **deliberation network**, which is an LAS decoder that attends to both the acoustic representations as well as the first-pass hypotheses, to generate the output. Let’s call this model (B).

The Generative Error Correction (GER) model (which this paper extends) is analogous to model (A), with the exception that it formulates the rescoring problem as a many-to-one sequence transduction task, as opposed to a simple re-ranking task.

The proposed model is similar to model (B), as is clear from equation (2). Its goal is to generate a new sequence given acoustic representations and an N-best list. In fact, the “early” and “mid” fusion methods in the paper are, in my opinion, exactly the same as a deliberation model, with the exception that the decoder is frozen instead of being jointly trained.

Situating late fusion is a little more complicated since the fusion only happens at the scoring stage, and it is quite confusing to me exactly how to think about it. This “late fusion” is, in a way, similar to shallow fusion methods in E2E ASR, with the exception that a GER model is used in place of the external LM. The other view is how the authors present it: GER with additional acoustic information.

While it is completely acceptable to use new terminology that is more in keeping with broader advances in LLMs, I think it would be beneficial to situate the proposed method in the context of the above ASR+LM strategies to make the paper more widely accessible. In summary, the use of LMs in ASR can be through either first-pass decoding or second-pass rescoring (this paper falls into the latter). Second pass rescoring may be modeled as re-ranking or sequence generation, and methods may or may not use the original acoustic information. This kind of taxonomy and categorization can be used to clarify the contribution of this work.

[1] Jelinek, Frederick. “Continuous speech recognition by statistical methods.” Proceedings of the IEEE 64 (1976): 532-556.

[2] Mohri, Mehryar et al. “Speech Recognition with Weighted Finite-State Transducers.” (2008).

[3] Ljolje, Andrej et al. “Efficient general lattice generation and rescoring.” EUROSPEECH (1999).

[4] Sak, Hasim et al. “On-the-fly lattice rescoring for real-time automatic speech recognition.” Interspeech (2010).

[5] Chorowski, Jan and Navdeep Jaitly. “Towards Better Decoding and Language Model Integration in Sequence to Sequence Models.” Interspeech (2016).

[6] Sriram, Anuroop et al. “Cold Fusion: Training Seq2Seq Models Together with Language Models.” Interspeech (2017).

[7] Sainath, Tara N. et al. “Two-Pass End-to-End Speech Recognition.” ArXiv abs/1908.10992 (2019): n. pag.

### Evaluation problems

If we accept the above categorization of models, the paper can be viewed in 2 ways: (i) shallow fusion of ASR and GER, and (ii) GER with acoustic information. As such, the proposed method should be compared against the following baselines: (1) end-to-end ASR, (2) shallow fusion of ASR and LLM, and (3) GER-only. The authors have compared against (1) and (3), but not against (2), which may be important to show how using GER improves compared to simply using a LLM with shallow fusion.

This is important because training the GER itself requires generating N-best hypotheses for the whole training data, which is computationally expensive, and it can only be used in an offline manner, while simple shallow fusion can be done on-the-fly. As such, we expect to see significant WER gains when using the GER module.

Another concern, which is a common issue when using LLMs, is the following: *How can we ensure that the test data was not seen by LLAMA?* In particular, the authors show in Table 3 that we get a large improvement on CHiME-4 when using the proposed method. CHiME-4 is essentially WSJ read outdoors, and WSJ consists of text from newspapers, which are in the public domain. It would be more useful to conduct experiments on a closed dataset, such as CHiME-5, which is less likely to have been memorized by LLAMA.

### Other minor comments

1. Some sections of the paper use a lot of flowery language which can be avoided. For example, in the last paragraph in Section 1, extraneous terms such as “a pioneering UADF technique”, “adroitly allocates”, and “conspicuously outperforming” can be made concise. In general, authors should stick to reporting rather than embellishing.
2. In Section 2 (paragraph 1), the authors mention that language models have been used in ASR for two decades, but the oldest citation is only from 2019. I think they should conduct a more thorough literature review in order to perform correct credit attribution.
3. The use of the terms “early”, “mid”, and “late” fusion in the paper may be confusing for readers familiar with ASR methods such as shallow, cold, and deep fusion. The authors should consider either changing *fusion* to another word, or making the difference explicit in the paper.

**Questions:**

Some of the proposed methods are specific to particular model choices. For example, (i) “early fusion” assumes that we are using Wav2Vec 2.0 as the encoder, and (ii) late fusion would only work with an encoder-decoder style model. The dominant ASR modeling strategy in the industry is conformer-transducers, which do not have such speech tokens, and their logit space is 3-dimensional (where vocabulary dimension also includes a blank token). Have the authors considered how the proposed method would work with such models?

---

> ### Author Response · Authors · 2023-11-15
> **Response for Reviewer 3ovr**
>
> We sincerely appreciate Reviewer 3ovr provides professional comments, which is quite constructive for improving this work. Now we respond to your concerns and questions:
>
> - **Q1: A more thorough literature review.**
>
>     Thank you for pointing out our oversight. We have modified this part and please see it in the new revision paper.
>
> - **Q2: The language style in Introduction.**
>
>     Thanks for your comment. We have modified our introduction according to your suggestion.
>
> - **Q3: The name of proposed fusion methods.**
>
>     We fully understand your concern. According to your suggestion, we add a paragraph in Section 3.1 to illustrate the relationship to
>     previous works, which includes the shallow fusion, cold fusion, and deep fusion you metioned.
>
> - **Q4: The evaluation problem of shallow fusion.**
>
>     We apologize for the misunderstanding caused. The “static fusion" baseline in our experiment can be viewed as a kind of “shallow
>     fusion” approach. In static fusion, we conduct the grid search on a small unseen validation (200 utterances) to find the suitable
>     weight to balance the weights of LLM and ASR systems. This process is akin to searching for the optimal hyper-parameters of LM’s
>     weight in shallow fusion.
>
>     Additionally, we have some other thoughts on your point about the comparison with shallow fusion. As you said “simple shallow
>     fusion can be done on-the-fly”, the foundation is that we have a well-trained ASR model and a language model (usually trained with
>     in-domain data). However, since UADF does not need any joint training process, it also can be done on-the-fly based on a well-
>     trained ASR model and GER model. In other words, it just replaces an in-domain LM with an in-domain LLM to independently
>     perform GER. Furthermore, because the GER utilizes the LoRA-tuning, the experiment of a single dataset can be completed in less
>     than one day with a single GPU (Nvidia A40 or 3090). Although this still requires more time than training an in-domain LM, GER is
>     more capable of independently predicting transcription, also resulting in a significant improvement to the ASR performance.
>
> - **Q5: Application in Conformer-transducers models.**
>
>     Thank you for bringing up this topic. We think that you provide a potential solution for a practical scenario: the conformer-transducer
>     model provides prompt response for users, and the LLMs perform error correction for high-quality results. To integrate them, we
>     think an asynchronous decoding strategy should be proposed, where LLM and transducer need to launch two separate decoding
>     iterations. When the transducer decoder predicts a “blank” token, it should skip the fusion process. Otherwise, the logits are passed
>     into LLM’s decoding iteration for dynamic fusion. However, under this setting, a primary challenge in asynchronous decoding we can
>     identify is the mismatch in length. A further alignment operation is required to ensure the temporal consistency between the logits
>     from two decoders.
>
> - **Q6: How to ensure ASR corpus is unseen for LLaMA.**
>
>     Thanks for your question. This concern has been discussed in Hyporadise paper, we report their observations here: 1) They
>     randomly write some customized sentences with errors that are unseen for LLM, while these errors can be corrected. 2) They
>     employ the T5 model (trained with publicly available C4 dataset) as the error correction backbone model, and it still works on
>     different datasets.
>     To reconfirm this issue, we reviewed the pre-training dataset introduced in LLaMA paper [1], and also checked each subset of
>     training data. No traces of WSJ’s transcription were found in these datasets. Furthermore, considering that WSJ is a paid dataset, its
>     transcription is typically not used for pre-training of LLMs.
>
> - **Reference**
>
>     [1] Touvron, Hugo, et al. "Llama: Open and efficient foundation language models." arXiv preprint arXiv:2302.13971 (2023).
>
>
> Thanks again for your time and patience. We hope our explanation can respond to your concern.

---

> > ### Comment · Reviewer_3ovr · 2023-11-20
> > **Increase score to weak accept**
> >
> > Thanks for your response to my comments. With the edited manuscript, my concerns about literature review, flowery language in the text, and the possibility for test set leakage in the LLM are alleviated.
> >
> > Regarding the other concerns, here are my additional comments:
> >
> > Q4: This is a nice equivalence, and it would be useful to point it out explicitly in the results section.
> > Q5: Sounds like a interesting possibility for future work --- I'll let this pass for this review.
> >
> > Based on the above, I am happy to increase my score to a weak accept.

---

### Official Review · Reviewer_1EWw · 2023-10-30

**Soundness:** 3 good
**Presentation:** 3 good
**Contribution:** 3 good
**Rating:** 6
**Confidence:** 3

**Summary:**

This paper introduces a late fusion strategy called "UADF" that combines the modalities of LLM and ASR to enhance generative error correction. The research demonstrates promising results across multiple datasets.

**Strengths:**

1. Integrating acoustic information with LLM is a promising strategy that enhances the capabilities and potential of LLMs in ASR tasks. This approach leverages the strengths of both acoustic processing and advanced language modeling, providing a more robust framework for GER.
2. The paper conducts a comprehensive examination of different fusion strategies, delving into detailed analyses of each approach.
3. The proposed UASF yields promising results across datasets with varying conditions, including ASR and VASR.

**Weaknesses:**

I believe it's not entirely fair to make comparisons with GER given the use of acoustic information.  It would be better to add some comparisons of the results with LLM scoring.

**Questions:**

1. What distinguishes the proposed method from shallow fusion using LLM?
2. In equation 5, do $f^{llm}$ and $f^{asr}$ need to be of the same length? If they do, how to ensure this?

---

> ### Author Response · Authors · 2023-11-15
> **Response for Reviewer 1EWw**
>
> We sincerely appreciate Reviewer 1EWw for considering that this work is comprehensive and provides a robust framework. Now we respond to your concerns and questions:
>
> - **Q1: Comparison with GER approach and other LLM rescoring baseline**
>
>     We agree that UADF uses more acoustic information than GER. However, for LLM, this kind of acoustic information is
>     unrecognizable, or even harmful due to the large modality gap, according to the results of early fusion. Our work aims to leverage
>     this acoustic information to improve the ASR results. Therefore, we compared the proposed UADF with early fusion and mid fusion
>     that also utilize the acoustic information.
>
>     For the rescoring baseline, as GER has surpassed the upper bound of the rescoring-based method (as oracle $o_{nb}$ shown in
>     Hyporadise paper), so we directly compare with the result of GER. We add a further LM rescoring baseline and oracle information in
>     Appendix A.1 Table 6.
>
> - **Q2: The length of $f^{llm}$ and $f^{asr}$**
>
>     We wonder the “length” you asked is the length of distribution or decoding sequence, but both of them need to have the same
>     length. For distribution, they are the vocabulary size 1D embedding, since our ASR model has been aligned with LLM. For sequence
>     length, as it is auto-regressive decoding, so they predict the current token together including the <eos> (end of sentence), so the
>     sequence length remains consistent throughout.
>
> - **Q3: The difference with shallow fusion.**
>
>     Thanks for your question. We add a paragraph to illustrate it in Section 3.1. UADF performs fusion at the same place as shallow
>     fusion. However, UADF adopts a dynamic fusion after analyzing the uncertainty of LLMs prediction, while shallow fusion employ a
>     static weight (our static fusion baseline is a kind of shallow fusion). Furthermore, in typical shallow fusion, LM probability usually is
>     assigned a small weight since it is viewed as auxiliary information. In UADF, both systems can independently predict transcription
>     and LLM plays a main role since GER shows better WER results than ASR.

---

> > ### Comment · Reviewer_1EWw · 2023-11-23
> >
> > I appreciate the author's response. I don't have any further questions, and I would like to maintain the current score.

---

### Official Review · Reviewer_2SYt · 2023-11-06

**Soundness:** 2 fair
**Presentation:** 2 fair
**Contribution:** 3 good
**Rating:** 6
**Confidence:** 4

**Summary:**

Recent studies showed that large language models (LLMs) can be successfully used to map the N-best hypotheses list generated by an ASR system to the predicted output transcription to improve ASR accuracy. But LLM is not trained with acoustic information. This paper proposed a method named Uncertainty Aware Dynamic Fusion (UADF) to infuse acoustic information to the auto-regressive decoding process with ASR decoder and LLM predictions. Specifically, it uses late fusion to combine the probability output by ASR decoder and LLM to predict the recognition results. In this process, temperature scales are applied to the logits of ASR decoder and LLM to match up the model confidence and recognition accuracy. Besides, the ASR and LLM combination weights are decided by LLM entropy so when LLM’s uncertainty is larger, more compensation from the ASR model will be used to decide the recognition results. The proposed method showed obvious WER reduction for several ASR tasks.

**Strengths:**

The paper proposed several methods applied to the fusion of ASR decoder and LLM output to improve performance. The paper compared the proposed method with several existing methods and did some ablation study to analyze the effectiveness of the proposed method. It also did experiments to prove the generalization of the proposed method. All the results showed the effectiveness of the proposed method.

**Weaknesses:**

•	Some contents are not accurate or clear.
o	In eq. (9), the entropy of LLM output is not accurate since for a discrete distribution, the entropy should be the sum of -p(y_t,i)logp(y_t,i).
o	In figure 2, it’s clear the probability of “all” from ASR is high. But it didn’t show where is the probability of “all” from LLM ? And what’s the meaning to show the blue cross in the left bottom part and the green cross in the right bottom part?
o	In the experiments part, for all late fusion experiments, the ASR model needs to be retrained with the training data. But it’s not clear whether all the training data are used to get a unified model or for each task (WSJ, ATIS, Chime-4), the model is updated only with the training data from this task.
o	The results in table 2 showed that without calibration, the proposed UADF did not perform better than the static method (WER 1.39 vs. 1.36). This means calibration is crucial for UADF, so the tuning of temperature t1 and t2 becomes a key to make it work. But the paper didn’t give enough information on these two parameters, such as what’s the optimal value, should we tune them for different tasks and so on.

**Questions:**

1.	Do we have the WER of the top 1 hypothesis in HyPoradise dataset for different tasks? These values may not be useful to validate the proposed method. But it will make the readers understand better why we need to use LLM to refine the N Hypotheses.
2.	As noted above, tuning of temperature t1 and t2 becomes a key point to make the proposed method work. The questions about this are:
a.	Did the author observe the same conclusion for other tasks (WSJ,  Chime-4, LRS3)?
b.	Do we need to tune these values for different tasks, or we could get them with one task’s data and applied them to other tasks.
c.	How much minimum data should we use to tune these values?
d.	What’s the optimal value for these two parameters for the tasks in this paper?
3.	In eq(10), the weight of LM and ASR didn’t sum up to 1.0. Will it matter since this means for different t, the range of P(y_t) will be different.
4.	It’s noted that “beta” in eq. (10) is set to 0.5 for the experiments. It didn’t show what’s the performance is if we change it to other values. Will the results be sensitive to this value?

---

> ### Author Response · Authors · 2023-11-15
> **Response for Reviewer 2SYt**
>
> We sincerely appreciate Reviewer 2SYt for considering that this work is effective, and your  suggestion is constructive. Now we respond to your concerns and questions:
>
>  -  **Q1: Explanation of Figure 2.**
>
>     We are sorry for the unclear presentation. In Figure 2, we only show the top-2 candidates from ASR and LLM, where the probability of
>     “all” is too low (close to 0) to be presented. For UADF, we present the top-3 candidates after fusion. We have revised the caption to
>     illustrate it.
>
> - **Q2: Training data for ASR model in late fusion.**
>
>     We retrain the ASR model as the tokenizer mismatch between ASR model and LLM. In other words, if Whisper has the same
>     decoding space with LLaMA, this step is not necessary as Whisper can perform as a general ASR engine. For the dataset you
>     mentioned, we employ a common Whisper encoder and train separated decoder (very small) for each dataset. These decoders
>     share the same decoding space with LLaMA.
>
> - **Q3: The top-1 hypothesis in HyPoradise dataset for different datasets.**
>
>     We have included it in the Appendix A.1 Table 6. Furthermore, we provide a LM rerank baseline and oracle performance for
>     comparison.
>
> - **Q4: The importance of tuning t1 and t2.**
>
>     Thank you for bringing up this topic. Firstly, as we mentioned in this paper, we find t1 and t2 on each dataset with binary search
>     algorithms on a small validation set including only 200 unseen utterances (they can provide thousands of token examples). This
>     search strategy is also used in classification tasks [1]. However, compared with classification tasks, we have indeed found out that
>     the impact of t1 and t2 are intricate to the WER performance. Now we provide some empirical conclusion to illustrate the impact of
>     t1 and t2 according our experiment:
>
>     (1) t1 acts on the LLM to alleviate its overconfidence problem. When WER is low, LLM tends to always output a confidence very
>     close to 1, which is difficult to be corrected if it’s a wrong prediction. Accordingly, t1 would reduce the peak value of the distribution.
>     (2) t2 acts on the ASR model that also exhibits an overconfidence phenomenon. Meanwhile, the calibration of ASR results in a
>     smoother probability distribution, which encourages more potential candidates to LLMs according to E.q.(10).
>     (3) In general, the benefits of t1 and t2 vary from different dataset due to their different WER levels. So we think the best solution is
>     employing a small validation set to find them for each dataset. Moreover, static fusion baseline also requires a validation set to find a
>     suitable weight (grid search).
>
>  - **Q5: In eq(10), the weight of LM and ASR didn’t sum up to 1.0.**
>
>     Many thanks for your comments. We understand your concern, and we have added a further softmax function for each step. But
>     please allow us to explain that in E.q.(10) the token index with highest value would be selected as results for each step, so this
>     operation would not affect the decoding strategy.
>
>  - **Q6:  The value of “beta”.**
>     We set beta as a fixed value of 0.5 in the ASR experiments that avoids introducing too many hyper-parameters. In practice, the
>     results are not sensitive to beta when it varies from 0.3 to 0.5 on ATIS, WSJ, and ChiME, as the GER performs better than ASR.
>     However, on other datasets, when two systems achieve comparable WER results on validation set (e.g. low SNR condition in AVSR
>     task), we can reduce the value of beta to improve the importance of the second model in E.q.(10).
>
>
>
> - **Reference**
>
>     [1] Kumar A, Ma T, Liang P, et al. Calibrated ensembles can mitigate accuracy tradeoffs under distribution shift[C]//Uncertainty in
>     Artificial Intelligence. PMLR, 2022: 1041-1051.

---

### Official Review · Reviewer_kWJM · 2023-11-07

**Soundness:** 3 good
**Presentation:** 3 good
**Contribution:** 2 fair
**Rating:** 5
**Confidence:** 3

**Summary:**

The authors present an approach they describe as Uncertainty Aware Dynamic Fusion (UADF) that 1. calibrates the LLM score at the token level accounting for the over-confidence of the neural network when doing auto-regressive decoding, 2. a time-changing uncertainty uncertainty that dynamically adjusts the decision-level fusion of text-based LLM and ASR scores. The results show significant improvement over strong baseline models such as Whisper and wav2vec or Hubert on a variety of tasks. Further the authors demonstrate that the results can be applied in an AVSR task with the dynamic uncertainty improving results in noise over a static fusion.

**Strengths:**

Overall the work is well motivated and demonstrates strong empirical results.

**Weaknesses:**

The point about calibration in using LLMs for rescoring of ASR hypotheses may be novel, however both calibration of hybrid or neural network ASR model scores and the use of strong language models combined with ASR scores are not novel approaches and have been widely used within speech.

The paper seems to desire establishing the term "modality laziness" as a technical way of describing when multimodal system underachieve compared to a unimodal system; in this case I would describe this as a simply a poorly designed system. There are multiple papers in the literature that describe effective ways of combining audio and visual modalities to yield an improve AVSR system over any single modality. Look for the term "modality drop out" and work on audio-visual conformers or AV-hubert.

Applying the entropy of the LLM predictive posterior eqn (9)  as a weighting factor for the dynamic fusion neatly correlates with the desired property of: when the LLM is more certain, then the ASR weight of the ASR can be less, and when the LLM is uncertain, then the ASR weight is higher.  However, it doesn't stem from the statistical approach of ASR where
Y_T = argmax Y_n p(X|Y_n; M_am) P(Y_n; M_lm ) where the first term in the argmax is the acoustic score and the second term the text-based LLM score, which could be computed with a LLM. In this light, without a Bayesian motivation, one might consider UADF as a hueristic method of combining LLM scores with ASR.

Lastly, the results here are all conducted on speech recognition tasks. Its unclear if such an approach can be applied to non-speech tasks or is of interest to the broader ICLR community.

**Questions:**

Can you re-frame UADF in a Bayesian framework?

---

> ### Author Response · Authors · 2023-11-15
> **Response for Reviewer kWJM**
>
> We sincerely appreciate Reviewer kWJM for considering that this work is well motivated and demonstrates strong empirical results. Now we respond to your concerns and questions:
>
> - **Q1: Discussion on modality laziness problem**
>
>     We agree with your comment about the audio-visual combining approach in AVSR system. However, we need to clarify that our
>     method is complementary to those approaches, e.g., modality dropout. Specifically, take the AV-HuBERT, you mentioned, as an
>     example, the modality laziness still exists: When the SNR is low, bimodal AV-HuBERT performs worse than visual-only modality
>     according to [1]. Therefore, we argue that modality laziness can not be completely avoided by representation learning approaches in
>     AVSR, and our proposed method can work together with modality dropout-like methods to further mitigate this problem and improve
>     the performance.
>
>     With respect to our main task: The motivation of our paper is to explore a suitable strategy to make LLM leverage acoustic
>     information for GER, and modality laziness is a problem when we fuse acoustic information to LLMs - refer to the result with early
>     fusion. This is intuitive because LLMs are very proficient in language but unfamiliar with speech. To this end, we propose the UADF
>     approach that performs a dynamic fusion for each decision step of token prediction, which avoids the joint training of two systems.
>
> - **Q2: Extent on Non-speech tasks to the broader ICLR community.**
>
>     Thanks for your suggestion. Our motivation is to make the LLMs leverage acoustic information, and the challenge mainly stems from
>     the large modality gap between speech and text. Therefore, we evaluate the UADF on ASR and AVSR tasks. Furthermore, there is a
>     potential application case is image captioning, where an image-to-text model is designed to describe the input image with natural
>     language. Since the mainstream image caption models are auto-regressive decoding, we argue that an external language model can
>     be dynamically fused to improve the quality of predicted captions.
>
> - **Q3: Re-frame UADF in a Bayesian framework.**
>
>     Eq. (10) can be thought of as a system combination scheme carried out in R^V space, and it is not related with the well-known plug-
>     in MAP decision rule used in speech decoding (plug-in maximum a posteriori decoder), where a likelihood (P(X|Y) and a prior
>     probability P(Y) are combined to obtain the best sequence Y*.
>
>     In the present work, we decided to handle uncertainty leveraging entropy of the posterior distribution given in Eq. (9) for the reasons
>     discussed in the paper. The proposed system combination scheme could eventually be re-formulated and casted into a Bayesian
>     framework following, for example, [2], in future work.
>
> **Reference**
>
>     [1]Chen, Chen, et al. "Leveraging modality-specific representations for audio-visual speech recognition via reinforcement learning." Proceedings of the AAAI Conference on Artificial Intelligence. Vol. 37. No. 11. 2023.
>     [2]Kim, Hyun-Chul, and Zoubin Ghahramani. "Bayesian classifier combination." Artificial Intelligence and Statistics. PMLR, 2012. Available from https://proceedings.mlr.press/v22/kim12.html.

---

> > ### Comment · Reviewer_kWJM · 2023-11-20
> >
> > Thank you for the reply. Given the responses I wish to keep my score unchanged. Detailed responses below.
> >
> > * Re Q1 response:
> > This is an unfair comparison in [1]: if you've trained your model only on clean data, then you've taught it to expect clean data. The fairer comparison is to provide noise at similar levels to what you expect to test it in, which is called multi-style or multi-condition training. Of course, you may test with an unseen type of noise, but the level of the added noise in training can be chosen with a distribution that at least exposes the model to the levels you are testing. Without using multi-style training, which is easily applied, the baseline is much weaker than it could be making any compensation technique appear better than it would against a stronger widely used baseline in the field. Of course for the actual paper under review, the Whisper model is a given and the experiments cannot be made to change its training.
> >
> > * Re Q2 response:
> > I agree with the authors that the technique presented could be applied to other domains outside of speech, however potential is different than actual experiments.
> >
> > * Re Q3 response:
> > Interesting, reference! Would be neat to see this reformulated in the Bayesian framework in [2].

---

> ### Author Response · Authors · 2023-11-20
> **Response for Reviewer kWJM**
>
> Thanks again for your feedback. We fully understand your suggestion in "Re Q1 response", but we think there is some misunderstanding about our work and [1]. We think the misunderstanding mainly stems from Table 1 in [1] where WER is extremely high when SNR=-15. However, we **do not add SNR=-15** in our paper, instead, our setting {−10, −5, 0, 5} exactly follows the AV-Hubert paper [2], which proposed a noise-robust AV-Hubert that contains the techniques you mentioned (e.g. dropout and adding training noise). They define these SNR levels to test the model performance and we think it is reasonable. Here we explain more details about your concern.
>
> - The AV-Hubert model used in [1] and our paper is trained by the "multi-condition training" approach you mentioned, as "Babble" noise **is included** in the training process and it is **seen noise**. More training details can be found in [2].
>
> - The modality laziness problem also can be found in the Table 2 ("Babble" column) of [2]. As [2] has included the dropout-like technique you mentioned, it shows that our method is complementary to these methods and can further improve the result without further labeled data.
>
> - Finally, besides the noisy condition, the proposed UADF also improves the performance in **clean** and **high SNR** conditions. In other words, UADF can alleviate the modality laziness in low SNR, but our contribution is not limited to it.
>
>
> [1] Chen, Chen, et al. "Leveraging modality-specific representations for audio-visual speech recognition via reinforcement learning." Proceedings of the AAAI Conference on Artificial Intelligence. Vol. 37. No. 11. 2023.
>
> [2] Shi B, Hsu W N, Mohamed A. Robust self-supervised audio-visual speech recognition[J]. arXiv preprint arXiv:2201.01763, 2022.
>
> We sincerely appreciate your time and effort. We look forward to discussing more with you on this issue.

---

### Comment · Area_Chair_mxPt · 2023-11-10
**reviewer-author discussions**

Dear All,

The reviewer-author discussion period will be from Nov. 10 to Nov. 22. For reviewers, please read the authors' responses and acknowledge it, respond to them early on in the discussion, and discuss points of disagreement. Thank you!

AC

---

### Author Response · Authors · 2023-11-15
**General Response for All Reviewers**

We express our gratitude to all reviewers for their insightful and constructive feedback. In response to the concerns raised, we have introduced the due changes, and our major modifications are summarized as follows:

- We enhanced the related work section to provide a more comprehensive literature review.
- A new paragraph has been added in Section 3.1 to elucidate the connection between the proposed method and existing fusion methodologies.
- We report more comparison information for UADF in Appendix A.1 Table 6.
- Further refinements have been made throughout the manuscript, including in the introduction, figure captions, and additional details in the Appendix, aligning with specific suggestions from the reviewers.

More details are explained in the comment for each reviewer.

---

### Meta-Review · Area_Chair_mxPt · 2023-12-05

**Metareview:**

The paper introduces an innovative approach named Uncertainty Aware Dynamic Fusion (UADF), designed to enhance the auto-regressive decoding process by dynamically assimilating the information from the acoustic modality during the auto-regressive decoding process by fusing ASR and LLM score. Specifically, it firstly calibrates the ASR and LLM scores at the token level accounting for the over-confidence of the neural network. Subsequently, the combination weights for ASR and LLM are dynamically determined based on the entropy of the LLM, ensuring that when the LLM's uncertainty is higher, the ASR model contributes more to the recognition results. The experimental results demonstrate a noteworthy enhancement across various speech recognition tasks.

The paper confronts several concerns:

Limited Novelty of Proposed Method: The novelty of the proposed method appears to be relatively weak. Score calibration through adjusting the temperature of logits in neural network models is a widely adopted technique across various machine learning tasks. Furthermore, the exploration of fusion-based methods to amalgamate scores from acoustic and language models in speech recognition is a well-explored domain within ASR. The paper should strive to articulate more precisely what sets its approach apart from existing methodologies in these areas.

Task-Specific Robustness Issues: The proposed method exhibits a lack of robustness across diverse tasks. Calibration temperatures (t1 and t2) necessitate task-specific tuning to outperform static fusion (i.e., standard shallow fusion). While the author suggests that only 200 utterances are required for tuning, the practicality of accessing such dev sets before system deployment may be challenging. Addressing this limitation is crucial for the broader applicability of the proposed method.

Questionable Baseline for Comparison: The choice of baseline for comparison raises questions. UADF essentially refines the shallow fusion method, while the Grounding Error Rate (GER) represents a distinct approach involving the training of a mapping from N-best to ground truth. Considering GER as the baseline seems unconvincing, and comparing against shallow fusion might be more appropriate. However, the fact that the proposed method surpasses shallow fusion only through the fine-tuning of calibration temperatures for different tasks diminishes its practical value, especially in real-world applications. A more thoughtful selection of baselines is essential for a comprehensive and valid evaluation of the proposed method's performance.

Limited scope for ICLR audience: The results here are all conducted on ASR tasks. Its unclear if such an approach can be applied to non-speech tasks or is of interest to the broader ICLR community, although the authors hypothesize in the rebuttal it should be applied to the image caption task. Verifying the proposed method in a non-speecht task will make the paper stronger.

**Justification For Why Not Higher Score:**

The paper has several weaknesses in all these areas: novelty, experiments, and scope, as noted above.

**Justification For Why Not Lower Score:**

This paper definitely has its contribution, especially in the time that speech community begins to look into how to integrate LLM. This clearly benefits the speech community, and is also valuabe to the ICLR audience on the multimodal integration of LLM.

---

### Decision · Program_Chairs · 2024-01-16

Accept (poster)